# Do Ectomycorrhizal Trees Select Ectomycorrhizal Fungi That Enhance Phosphorus Uptake under Nitrogen Enrichment?

**Thomas W. Kuyper** [1,*] **and Laura M. Suz** [2]

1 Soil Biology Group, Wageningen University and Research, P.O. Box 47, 6700 AA Wageningen, The Netherlands
2 Royal Botanic Gardens, Kew TW9 3DS, UK
* Correspondence: thom.kuyper@wur.nl

**Abstract:** Globally, forests are impacted by atmospheric nitrogen (N) deposition, affecting their structure and functioning above and below ground. All trees form mutualistic root symbioses with mycorrhizal fungi. Of the two kinds of mycorrhizal symbioses of trees, the ectomycorrhizal (EcM) symbiosis is much more sensitive to N enrichment than the arbuscular mycorrhizal (AM) symbiosis. Due to increasing N deposition, significant declines in the richness and abundance of EcM fungal species and shifts in community composition and functional traits have been recorded. Under increasing N deposition, ectomycorrhizal forests usually show enhanced foliar mass fractions of N, reduced foliar mass fractions of phosphorus (P), and, consequently, an increasing imbalance in the foliar N:P stoichiometry, ultimately impacting tree performance. The question has been raised of whether, under conditions of high N deposition, EcM trees can select EcM fungi that are both tolerant to high N availability and efficient in the acquisition of P, which could to some extent mitigate the negative impact of N deposition on nutrient balances. Here we evaluate the literature for mechanisms through which certain EcM fungi could increase P acquisition under increased N loading. We find very little evidence that under N enrichment, EcM fungi that have on average higher P efficiency might be selected and thereby prevent or delay tree N:P imbalances. However, methodological issues in some of these studies make it imperative to treat this conclusion with caution. Considering the importance of avoiding tree N:P disbalances under N enrichment and the need to restore EcM forests that have suffered from long-term excess N loading, further research into this question is urgently required.

**Keywords:** ectomycorrhiza; arbuscular mycorrhiza; nitrogen; phosphatase; phosphorus; stoichiometry; trees





## 1. Introduction

Human-induced environmental change has a large impact on the functioning of forest ecosystems. Among them, climate change and nitrogen (N) deposition are major factors that change the composition of tree species and the functioning of forest ecosystems. Trees associate with either ectomycorrhizal (EcM) or arbuscular mycorrhizal (AM) fungi in a mutualistic symbiosis [1], where the tree provides carbohydrates to the fungus and the fungus provides limiting soil resources, including nutrients like N and phosphorus (P). It is generally accepted that EcM forests, at least in temperate and boreal regions, are characterized by a relatively closed organic-nutrient economy, whereas AM forests are characterized by a more open inorganic-nutrient economy [2]. Ectomycorrhizal forests are also more important in young soils under N-limiting conditions, whereas AM forests are more important in old, weathered soils under P-limiting conditions. Anthropogenic deposition of either reduced (ammonium; $NH_4^+$) or oxidized (nitrate; $NO_3^-$) N forms adds mineral N to the forest floor and may result in a shift from a more closed to a more open N-cycle, favoring AM trees over EcM trees. This shift in tree species composition towards AM tree dominance, mainly due to higher seedling establishment of AM trees than EcM

trees at increased N availability, has been documented for forests in the United States of America [3,4].

As a consequence of N deposition, plant growth and forest productivity initially increase. Foliar N mass fractions (concentrations) also increase, whereas P mass fractions do not change or even frequently decrease. Increasing N mass fractions and decreasing P mass fractions cause stoichiometric imbalances in the foliar N:P ratio. In a meta-analysis, Mao et al. [5] noted that N enrichment increased foliar N mass fractions of trees (+15%), with a larger positive effect in boreal forests (+41%) than in temperate forests (+17%) and tropical forests (+9%). For foliar P mass fractions, they observed the opposite effect, with a decline in P mass fraction of 10% (boreal forests) or 6% (in temperate forests), with no effects in (sub-)tropical forests. Consequently, changes in foliar N:P ratios under the influence of N deposition were largest in boreal forests and smallest in (sub-)tropical forests, and these stoichiometric imbalances increased with higher annual N addition rates. The study did not provide data to assess whether there was a difference between AM and EcM forests; however, as most boreal forests are dominated by EcM trees, while most (sub-)tropical forests are dominated by AM trees, such a differential mycorrhizal guild effect is plausible. A meta-analysis by Ma et al. [6] indicated that N deposition resulted in a larger increase in foliar N mass fractions in EcM trees than in AM trees, whereas P mass fractions remained unchanged under N deposition for both tree mycorrhizal guilds, resulting in a larger increase in foliar N:P ratios in EcM trees than in AM trees and hence a higher risk of a stoichiometric imbalance for EcM trees. Foliar stoichiometric imbalances could further occur if N excess leads to N leaching and concomitant acidification. Due to acidification, the adsorption of both organic P and orthophosphate to mineral (iron-coated) surfaces increases, reducing P availability and further constraining the N:P balance under N enrichment. This increasing imbalance in foliar N:P ratios and P mass fractions below critical limits is deteriorating the mineral nutrition of trees, which ultimately constrains their ability to store carbon aboveground as shown for several ectomycorrhizal trees in Europe, such as *Quercus petraea* (Matt.)Liebl., *Fagus sylvatica* L., and *Pinus sylvestris* L. [7], as well as belowground, where most of the carbon is stored [8].

The high sensitivity of EcM fungi and, consequently, of EcM trees for N deposition has been repeatedly demonstrated [9–11]. High sensitivity is evident by changes in species composition and hence functional traits and reductions in the number of certain species, sporocarps, ectomycorrhizal root tips, and hyphal length density. There have also been reports of certain AM fungal species being highly sensitive to high levels of N deposition [12], but the negative effects on these fungi, and even more so on AM trees, are less severe than those of fungi and trees [9,10]. This differential sensitivity of EcM and AM forests is likely caused by the fact that EcM forests are characterized by a conservative, closed N cycle, whereas AM forests are characterized by a more acquisitive, open N cycle, which is therefore less disrupted by N deposition [2]. However, as N-tolerant AM fungi have been hypothesized to confer a lower P benefit to plants than N-sensitive AM fungi, N deposition could ultimately also constrain the ability of AM forests to store above-ground and below-ground carbon [12].

Individual EcM trees can be simultaneously colonized by a large number of EcM fungal species. Bahram et al. [13] reported the simultaneous occurrence of more than 100 EcM fungal species on a single *Populus tremula* L. tree. And while this number is currently a record, the occurrence of several dozens of different EcM fungal species in association with a single tree is common. These fungal species differ in a number of morphological and physiological traits, such as their soil exploration type, hydrophobic or hydrophilic properties of the mycelium, the $^{15}$N signature of the mycelium (which likely reflects the extent to which they acquire N through organic or mineral sources), or their nitrophobic or nitrotolerant character [14]. Functional differentiation in traits related to N acquisition can be beneficial from a tree perspective, as maintaining a higher EcM fungal species richness on its root system confers benefits for the tree while maintaining functional redundancy in case of potential diversity loss [15–17].

Could similar processes play a role in the P nutrition of EcM trees? Köhler et al. [18] demonstrated that higher EcM fungal species richness benefits P acquisition under drought and high temperature, but not under N loading. Their experimental study was executed at the very low end of species richness, with a realized species richness of less than three species on seedlings of *Fagus sylvatica*, so generalizations from this study are clearly premature. A subsequent study [19] demonstrated no differences between EcM fungal species, suggesting that intraspecific physiological adjustment was more important than interspecific niche differentiation.

The question that has hardly been addressed is whether species shifts in the EcM fungal community could to some extent compensate for the reduced ability to acquire P over time, as inferred by the link between shifts in EcM community composition and thresholds of foliar N:P ratios in temperate and boreal forests across Europe [9,10]. If an EcM tree can selectively reward EcM fungi that contribute more than the average amount to P acquisition, the trees' N-enrichment-driven nutrient imbalances may be mitigated. This possibility was suggested by Suz et al. [14], who stated that it is currently unclear if there are ecologically important, nitrotolerant P specialist EcM fungi. To the best of our knowledge, only one study has found support for this hypothesis; Kottke et al. [20] observed high P mass fractions in the sheath of the nitrotolerant *Imleria badia* (Fr.) Vizzini (formerly *Xerocomus badius* (Fr.) E.J. Gilb.), suggesting that this species is able to efficiently acquire and store P, with a potential to exchange this with its tree partner. Almeida et al. [21] noted that this species is stimulated under N enrichment but reduced under P enrichment, consistent with the possibility that it could alleviate P limitation to some extent. Although *I. badia* emerged as an indicator of high N deposition and showed plasticity in soil foraging traits, no data are available on whether this species' occurrence is associated with lowered N:P ratios [11].

In this review, we address this question and focus on the EcM plant and fungal mechanisms to maintain balanced N:P ratios and compare them to those of AM trees. We first discuss a predominant non-mycorrhizal mechanism, viz., the ability of EcM trees to reduce P losses by enhanced resorption, the retranslocation of nutrients from foliage to storage organs before leaf litter falls, under N deposition. Since resorbed P is partly stored in the fungal mantle [22], it could still be under some fungal control. Then we evaluate the literature for mechanisms through which certain EcM fungi could increase P acquisition under increased N loading. Finally, we look at ways in which EcM forests show a differential P speciation (i.e., the distribution of different P forms) in comparison to AM forests, and raise the question of whether this could delay the competitive replacement of EcM trees by AM trees. As the pertinent literature showed a number of potentially problematic methodological issues that were often not or only briefly discussed, we also reflected on methodological issues when testing the hypothesis that under N enrichment a guild of nitrotolerant, P efficient EcM fungi could prevent or delay a stoichiometric N:P imbalance [14].

## 2. Ectomycorrhizal, Fungal, and Plant Mechanisms to Prevent N:P Unbalances
### 2.1. Enhanced P Resorption

Ectomycorrhizal trees might cope with an increasingly unfavorable N:P balance as a consequence of N deposition and a concomitant switch from N-limited growth to P-limited growth by either reducing losses and/or increasing nutrient acquisition. Reducing losses could be achieved by increasing the efficiency of resorption before leaf or needle fall. Zhang et al. [23] studied differences in N and P resorption between AM and EcM trees, noting a large biome effect. Nitrogen resorption was higher for AM trees than for EcM trees in (sub-)tropical forests, whereas there were no differences between both guilds in temperate and boreal forests. However, a higher P resorption was recorded for EcM trees than for AM trees in boreal forests, whereas there were no significant differences between both guilds in temperate and (sub-)tropical forests. In contrast, in forests where both AM and EcM trees co-occur, N resorption was higher for EcM trees than for AM trees in temperate forests,

whereas the opposite pattern was noted for (sub-)tropical forests. For P resorption, there were no differences between the co-occurring guilds in both temperate and (sub-)tropical forests. Effects of N deposition on resorption efficiency were studied by You et al. [24], who noted that N deposition decreased N resorption but had no effect on P resorption. While their data imply that N deposition could create a disconnect between N and P resorption, **the lack of increase in P resorption efficiency suggests that the adaptive response of trees to N enrichment-driven stoichiometric imbalances is very limited.** This conclusion was confirmed by Deng et al. [25], who observed that alleviation of P deficiency due to N enrichment in a plantation of the EcM tree *Larix principis-rupprechtii* Mayr was caused by increased P uptake but not by enhanced P resorption efficiency.

*2.2. Increased P Acquisition*

On a global scale, a trade-off seems to exist between N resorption efficiency and N mineralization [26], with N mineralization increasing with temperature and precipitation while resorption decreases along that climatic gradient. It is not known whether the same trade-off exists for P or whether, under such conditions, mineralization (or direct acquisition) of organic P could be enhanced to maintain a stoichiometric balance. Phillips et al. [2] hypothesized that the difference between an open, inorganic cycle and more closed, organic nutrient cycle would apply equally to N and P; however, that is unlikely. Plants and fungi can acquire organic N from small N-containing molecules like amino acids, but the uptake of organic P has never been demonstrated. Only orthophosphate can be taken up by plants and fungi, implying that all organic P must pass through the orthophosphate funnel [27] before uptake. While many EcM fungi possess an organic-P transporter that is able to import several phospho-diesters, it is likely that this enzyme plays a role in the internal turnover of phospholipids, but its role in the uptake of external phospho-diesters has never been shown [28].

An implication of this orthophosphate funnel is that no niche differentiation for uptake from different organic-P sources is possible, contrary to what had been hypothesized before [29]. Tests of his hypothesis have still failed to confirm it. In a mixed AM-EcM plantation, Steidinger et al. [30] tested the ability of tropical montane trees to acquire P from organic sources. They noted that EcM trees did not perform better than AM trees in the presence of organic P sources and concluded that trees of both mycorrhizal guilds exploit the same forms of P. As their study demonstrated differences in phosphomonoesterase (commonly called phosphatase in many studies, and this term will also be used here), the question of whether EcM and AM fungi (and consequently, EcM and AM trees) differ in the amount of phosphatases produced and, if so, whether enzyme production can be upregulated after N enrichment in order to alleviate stoichiometric imbalances, is pertinent. As the production of these enzymes may cost the plants carbon (C), the question of whether EcM trees can selectively reward EcM fungi that either produce more enzymes or are able to upregulate enzyme production under N enrichment is relevant. Investment in phosphatases also incurs a N cost [31], and increased N availability could therefore result in increased phosphatase production.

Under normal conditions, microbes exhibit a balanced investment in enzymes related to C acquisition (hydrolytic and oxidative enzymes), N acquisition (chitinase and aminopeptidases), and P acquisition (phosphatases) [32], a pattern shown for both soils and sediments. Similar relations between N- and P-acquiring enzymes were established by Courty et al. [33] for EcM fungi; the activity of oxidative enzymes, which liberate organic N and P from organic matter, was also correlated with both N- and P-acquiring enzymes. However, coordination of enzymatic activities could be lost in cases of drastic changes in the supply of mineral nutrients, as in the case of N enrichment. Various meta-analyses on the relationship between N enrichment and phosphatase production have been published. Based on 34 studies Marklein and Houlton [34] noted that N fertilization increased phosphatase production on average by 22% in roots (including mycorrhizal roots) and 5% in soil; however, 26 out of 85 data points of soil assessment exhibited a reduction in phosphatase

activity. They did not subdivide their dataset per biome, nor did they test for differential effects between EcM and AM forests. A more recent meta-analysis [35] was based on 140 studies and 668 observations but was restricted to soil phosphatases. Overall and across all studies, N enrichment increased phosphatase activity by 13%; however, there was a significant difference between studies lasting less than five years (with an average positive effect of 28%) and studies lasting more than five years (with no effect of N enrichment at all). Their larger dataset allowed testing for the effects of separate biomes. For forests, they also noted a positive effect in short-term studies (less than five years) but no effect in studies that lasted longer than five years. Despite the fact that phosphatase production was no longer elevated in studies lasting five or more years, the authors suggested that the initial increase in phosphatase production would allow long-term alleviation of N-induced P limitation of plant productivity.

Two mechanisms could explain the attenuation of phosphatase production under long-term N enrichment. The initial positive response to the N addition to phosphatase production can increase P mineralization, resulting in increased P uptake and recycling of this newly available P. Under that model, there is a direct connection between initial stimulation of phosphatase activity and subsequent alleviation of P limitation [36]. However, an alternative mechanism would imply declines in the abundance and activity of both mycorrhizal and saprotrophic fungi and bacteria under N deposition, an effect that overrides the initial stimulation of phosphatase activity. Under that mechanism, there would be no alleviation, but rather an aggravation of P limitation over time. Differential effects of N enrichment on AM and EcM forests would be consistent with, though not an explicit test for, this alternative hypothesis. The study by Chen et al. [35] did not allow testing for differential effects between AM and EcM forests, and while they briefly discussed the role of mycorrhiza, they focused on the potential negative effect of N enrichment on AM fungi without discussing the much more significant effect of N enrichment on EcM fungi. They concluded that the potential of mycorrhizal fungi to alleviate P limitation with N enrichment was still unclear.

An explicit mycorrhizal focus on the impact of N enrichment on phosphatase production in forests was provided by Ma et al. [6], based on 116 papers. Their study confirmed the general positive response of N enrichment on phosphatase production (+8%) and a very strong difference between studies lasting three years or less (+49%) or studies lasting between three and ten years (−11%). Separation of AM and EcM forests showed that for AM forests there was a positive effect of N enrichment (+28%), whereas for EcM forests there was no effect (+4%, not significantly different from zero). Unfortunately, their Figure 8 provided the wrong image, as that figure suggests a stronger positive response in the case of EcM trees than of AM trees. They explained the larger effect in AM forests than in EcM forests by the fact that AM trees are usually P-limited (and so would benefit more from stimulation of phosphatase production), whereas EcM trees are more commonly N-limited [37]. This potential differential response of AM and EcM plants is supported by observations in mixed stands of *Larix principis-rupprechtii* with an understory of AM herbs, where N fertilization increased phosphatase activity with increasing N loading more in stands with a greater share of AM fungi, implying that there is a stronger adaptive response to N-induced nutrient imbalances in AM plants than in EcM plants [25].

Meta-analyses deal with fractional changes in relation to control treatments. A larger positive effect of N enrichment on phosphatase production in AM forests than in EcM forests does not imply that enzyme production as such is higher in AM systems. In fact, the evidence of whether phosphatase production by individual roots is higher for EcM than for AM roots is mixed. Antibus et al. [38] compared two EcM trees (*Betula alleghaniensis* Britt. and *Fagus grandifolia* Ehrh.) with one AM tree (*Acer rubrum* L.) and observed higher phosphatase and phytase activity for the EcM trees than for the AM tree. They also noted large differences in phosphatase production between two EcM morphotypes on *Betula*, with a smooth morphotype having somewhat higher phosphatase activity than a tomentose morphotype, although no taxonomic details were given about these morphotypes. Links

between EcM morphotype structure and the foraging strategy of the fungal mycelium have been proposed, with smooth morphotypes with little emanating mycelium exhibiting contact exploration types and morphotypes with abundant emanating mycelium exhibiting medium- or long-distance exploration types [39]. Exploration types have been connected to sensitivity to N enrichment, with contact types being more nitrotolerant, medium-fringe, and matt, and certain long-distance types being on average more nitrophobic [40]. However, exceptions occur, as in the case of the nitrotolerant *Imleria badia*, which exhibits a long-distance exploration type. However, unexpected fungal plasticity in exploration behavior [11] and differences in mycelial longevity [41] can lead to inconsistencies between soil exploration types and short-term mycelial responses to changes in nutrient availability.

Consistent with that study [38], Weand et al. [42] reported the lowest phosphatase activity in the AM tree *Acer saccharum* and the EcM tree *Quercus rubra* L., compared to three further EcM trees (*Tsuga canadensis* (L.) Carrière, *Fagus grandifolia*, and *Betula alleghaniensis*), and also recorded differential responses to N fertilization, with the largest increase in phosphatase activity for *A. saccharum*. Hirano et al. [43], in a study of both EcM and AM tropical forest trees, did not find differences between the two guilds in the production of phosphatase, phosphodiesterase, and phytase, or in their upregulation of P-acquiring enzymes after N addition.

The larger increase in phosphatase production in AM trees is surprising because genomic studies failed to demonstrate the presence of these genes in several AM fungi [44]. It has become evident that these phosphatases are not produced by the fungi but by hyphosphere bacteria that are closely associated with the hyphal wall [45]. Considering the too high foliar N:P ratios in European EcM trees under N deposition [7], **the inability of EcM trees and their associated fungi to increase phosphatase production under N enrichment may explain their poor regeneration and potential replacement by AM trees, as reported for the United States of America [3,4].**

Corrales et al. [46] noted that after nine years of N addition in a montane subtropical forest dominated by the EcM tree *Oreomunnea mexicana* (Standl.) J.-F. Leroy (Juglandacaeae), both phosphomonoesterase and phosphodiesterase activities in soil were significantly reduced by ca. 50%. The authors also noted a significant decline in the relative abundance of the EcM genus *Cortinarius* on the roots and a significant increase in the relative abundance of the EcM genera *Laccaria* and *Lactarius*. After correcting for the N treatment effect, the activity of both enzymes was positively correlated with the DNA amplicon abundance of *Cortinarius* and negatively with that of *Russula* and *Tomentella*. It should be noted that in a study by van der Linde et al. [11], two *Cortinarius* and one *Russula* species were classified as nitrophobic, while one *Tomentella* and three *Russula* species were classified as nitrotolerant, whereas in Lilleskov et al. [40], *Cortinarius* is classified as sensitive to N enrichment, whereas mixed responses have been reported for *Tomentella* and *Russula*. High phosphatase activity by members of the *tomentella-thelephora* lineage and low activity by members of the russuloid lineage (*Russula, Lactarius)* have also been observed [47,48]. However, across a gradient of declining P availability as a consequence of podzolization in forests of *Pseudotsuga menziesii* (Mirb.) Franco, the phosphatase production of *Cortinarius* was lower than that of *Tomentella*, *Russula,* and *Lactarius*. Along this gradient of declining P availability, as evidenced by increasing foliar N:P ratios, the correlation between leaf N:P and phosphatase was significant, suggesting that across this gradient, EcM species with high phosphatase capacities might be selected for [49]. A net decline in phosphatase production with increasing N loadings was reported for ectomycorrhizas of *Pinus thunbergii* Parl. in an urban environment in China [50].

Many studies have been executed on enzymatic production by roots, especially EcM roots, after the identity of the fungal species was assessed, usually through barcoding but occasionally through root tip morphology. Enzymes involved in C cycling (hydrolytic and oxidative enzymes), N cycling (involved in the degradation of proteins and chitin), and P cycling (phosphatases) are usually tested. Only a few studies tested the same species at different N levels or tested for plasticity in nutrient acquisition traits. On the

one hand, looking specifically at phosphatase production by excised root tips may have advantages because it allows linking enzymatic activity with fungal species identity. On the other hand, N enrichment may significantly reduce the number of EcM root tips and extraradical hyphal length, and if these changes are not accounted for, a biased picture may emerge. Measuring enzymatic activity over fixed volumes of soil could take such changes into account; however, enzymatic activity is then also determined by saprotrophic fungi and bacteria, and, due to enzyme sorption to mineral surfaces, carry-over effects may be equally important.

The study by Dunleavy and Mack [51] clearly demonstrates the potential risks of assessing enzymatic activity per unit (weight, surface area) of fine root. After 28 years of N and P fertilization in an arctic tundra dominated by the EcM *Betula nana* L., they observed lower activity of N-acquiring and oxidative enzymes but not of phosphatase. The main EcM fungal genera negatively affected by fertilization were *Cortinarius*, followed by *Russula* and *Lactarius*. There were no changes in phosphatase activity per root tip colonized by fungi of these genera, but because of the decrease in root tip abundance, the fertilization effect on the ecosystem level was negative. *Cortinarius* was the most sensitive to fertilization and had the highest phosphatase activity of the three genera, implying that there was a significant ecosystem effect.

A meta-analytical approach could also be complex when comparing between different EcM fungal species, because some species may exhibit low phosphatase activity but show upregulation after N addition (hence a positive response ratio in meta-analyses), while other species exhibit high activity without changes after N addition. This contrasting behavior was reported by Taniguchi et al. [52], who reported low phosphatase activity in *Suillus granulatus* (L.) Roussel and *Rhizopogon* spec. under low N but significant increases at high N and high phosphatase activity in two species of *Tomentella*. As root tip numbers declined very significantly after N addition for all four species, the effect of N on phosphatase activity per seedling was strongly negative, overriding any effect of fungal species identity.

### 2.3. Delaying Competitive Replacement by EcM Modification of Different Phosphorus Forms

The modification of P speciation, the distribution over different P forms, is one mechanism that could delay competitive replacement of EcM trees by AM trees. Differences in P cycling between EcM and AM stands have been reported [37] with, in agreement with the mycorrhiza-associated nutrient economy (MANE) framework [2], a more important organic-P cycle in EcM stands than in AM stands. More specifically, Rosling et al. [37] noted a higher phosphatase activity in the organic horizons of the EcM stands than of the AM stands and much smaller differences between both tree guilds in the mineral soil. Likely because of a lower pH in the EcM stands than in the AM stands, P sorption to mineral surfaces was significantly higher in the EcM stands, whereas soil solution P was higher in the AM stands. Qi et al. [53] also reported higher organic P in the organic layer with increasing EcM tree dominance in subtropical secondary forests but lower organic P in the mineral layer. However, in their study, there was no significant relationship between phosphatase activity and the relative abundance of EcM trees. As there was a significant positive relationship between citric acid extractable P (considered a proxy for P acquisition via organic acids) and the relative abundance of EcM trees, the authors hypothesized that organic acids played a larger role than enzymatic activity in conferring competitive benefits to EcM trees. After uptake and return to the soil as foliar litter, the amount of organic P in the litter layer would then increase.

The study by Weand et al. [42] did not find mycorrhizal effects on P speciation in the organic and mineral layer under trees of five different species (four EcM and one AM species), both under control and N-fertilized conditions, due to the fact that interspecific differences among the four EcM species were much larger than the mycorrhizal effect or the treatment effect. Lack of treatment effects could also have been caused by the fact that tree performance was already P-limited, judging from the foliar N:P ratios above 15 in four out of five species in the control, related to the very acidic conditions. In a subtropical

forest with the EcM tree *Castanopsis carlesii* (Hemsl.) Hataya, a decline in NaOH-extractable organic P was observed after N fertilization, which paralleled declines in microbial biomass P and biomass of both EcM and AM mycorrhizal fungi [54]. It is therefore not evident that the decline in NaOH-extractable organic P reflects increased use of a potentially available P source to meet increased tree P demand.

Enhanced weathering of mineral P (apatite) by so-called rock-eating EcM fungi under the influence of N deposition could also increase P availability [55]. However, it has yet not been established whether the weathering rates, due to proton excretion and organic-acid production, are sufficiently high for making an impact in the case of atmospheric N deposition, or whether EcM fungal-mediated weathering is especially relevant over (much) longer temporal scales [56].

Changes in P speciation could be driven by direct and indirect (acidification) effects on N deposition. In a mixed subtropical forest in China, acidification (the acid treatment consisting of $H_2SO_4$ and $HNO_3$) resulted in a larger fraction of P ending in the occluded pool, and a smaller fraction of P (both inorganic and organic) in the labile and moderately labile pools [57]. The authors noted the upregulation of phosphatase and a significant decline in microbial biomass P, a decline in AM fungal biomass, and no change in EcM fungal biomass. These data then suggest that direct effects of N deposition are more important than the indirect effects of N enrichment through acidification and increased adsorption of P to mineral surfaces, but these data also suggest that **the link between mycorrhizal symbiosis and the upregulation of enzymes involved in P acquisition from organic sources is unlikely to be direct.**

### 3. Limitations of These Studies

#### 3.1. Dealing with Multiple Causality

Replacement of EcM vegetation by AM vegetation was documented in two studies from the United States of America [3,4]. These changes have been interpreted as having been caused by increased N deposition; however, over large spatial scales, disentangling the effects of N deposition and climate change is complicated, as both are spatially auto-correlated. Therefore, attribution of effects may not always be straightforward, except for experimental studies on a small scale. The data by Jo et al. [4] did show that levels of N deposition were more important for the changes towards AM tree dominance than changes in mean annual temperature or mean annual precipitation. It should be added that climate change likely has a larger effect on N cycling, which is predominantly a biological process of immobilization and mineralization, than on P cycling, where the physicochemical process of adsorption and desorption outweighs the role of the biological P cycle. Deng et al. [25] showed a significant negative correlation between N resorption and N mineralization due to the strong impact of climate on N mineralization, which is then traded off against N resorption. However, the strong climatic effect on N resorption was absent for P resorption in a study by Jiang et al. [58], confirming the differential impacts of climate on N versus P cycling.

#### 3.2. Meta-Analyses

Estimates of effect sizes, as provided in this review, have often been derived from meta-analyses. While meta-analysis is a very powerful tool for aggregating effect sizes over a large number of studies, aggregation might hide contrasting effects within the total number of studies. Two meta-analyses [6,35] on changes in phosphatase production after N enrichment provide an example of this. Both studies reported a significant effect of the duration of the study, with short-term studies exhibiting a significant positive response and long-term studies exhibiting no response. The aggregated positive response in both studies (13% and 8%, respectively) is therefore a function of the relative balance between short-term and long-term experiments. However, there could be hidden factors in the factor of experimental duration, as the frequency of N addition (number of additions per year) is higher in short-term experiments than in long-term experiments, and the positive

effect of N on phosphatase increases with N loading frequency, the number of annual additions [35]. Ma et al. [6] also noted a significant difference between AM and EcM forests (28% versus 4%). Again, it is not clear whether these assessments are independent, as it could well be that studies on the effects of N deposition in temperate and boreal regions, where EcM forests prevail, have started earlier than similar studies in (sub-)tropical regions, where AM forests are more common. **Assessing relationships between seemingly, but not necessarily, independent variables (study duration or frequency of experimental treatment, and the nature of dominant mycorrhizal type) is still required to provide context for meta-analysis-based generalizations.**

*3.3. Enzymatic Assays*

Most studies of enzyme activity have been carried out in slurries rather than in situ. While easier from an experimental perspective, this approach assumes that for the conversion of organic phosphates into orthophosphate, enzyme content and activity are the limiting factors. This assumption has been contested for a long time, and already Tinker and Nye [59] have proposed that the hydrolysis of organic P is limited by the availability of organic P in the soil solution, due to the sorption of phosphate to mineral surfaces, rather than by enzyme activity. Jarosch et al. [60] confirmed that substrate availability was the limiting factor for phosphate monoesters but not for phosphate diesters. As most studies reviewed here assessed phosphomonoesterase, its activity should be studied in conjunction with mechanisms of organic P desorption via carboxylates. The combined action of carboxylates and phosphatases might explain why, out of 16 species of EcM fungi tested, several species produced phosphatase in a medium with apatite but without organic P [61], potentially suggesting a joint regulatory mechanism for both classes of compounds acting jointly in P acquisition.

Surprisingly, Jarosch et al. [60] suggested that phytate hydrolysis may be enzyme-co-limited, in addition to substrate availability, because phytate is more strongly stabilized on mineral surfaces than other organic-P sources (and thus the most abundant organic-P form in soils). Unfortunately, there is a dearth of studies on phytase production by EcM and AM fungi, and it is frequently incorrectly assumed that phosphomonoesterases can hydrolyze P from phytate. However, Nannipieri et al. [62] reported that phosphomonoesterases cannot hydrolyze phytate but only lower-order inositol phosphates. Therefore, a link between phosphatase production and P acquisition from the most recalcitrant organic-P sources is indirect at best. Current evidence indicates that phytate is hardly a usable P source for EcM plants [63], and that only in the presence of phytase-producing bacteria and protozoan bacterial grazers phytate-P can be delivered to EcM plants [64]. The only exception was a study by Liu et al. [65], who noted increased seedling biomass after addition of phytic acid in EcM trees but not in AM trees. Somewhat surprisingly, the addition of a simple phosphate monoester was a less beneficial P source for these EcM trees than phytic acid, even though the enzymatic hydrolysis of phytic acid through dephosphorylation ultimately generates these lower-order inositol phosphates. Hirano et al. [43] did not demonstrate higher phytase activity for EcM trees than for AM trees in a comparable rainforest.

Apart from a potential problem with the differential availability of organic P in soils and in slurries where enzyme activity is assessed, the enzymes themselves could also be adsorbed to mineral surfaces. This topic has been little investigated. Kedi et al. [66] studied the pH-dependent adsorption of three phosphatases (not further specified) of two EcM fungi (*Suillus collinitus* (Fr.) O. Kuntze and *Hebeloma cylindrosporum* Romagn.) on two tropical soils and found high variability in both their affinity for different soils and their pH-dependent behavior, precluding any generalization. Adsorption of phosphatases on mineral surfaces could generate large carry-over effects, where changes in a larger residual enzymatic pool (rather than direct up- or downregulation of enzyme activity due to treatment effects) drive changes in enzymatic activity after N enrichment. This phenomenon could partly explain reports of contrasting responses of microbial biomass and enzyme activity [54,57].

Finally, there are risks involved when working with excised roots for enzymatic assays. Excision and subsequent removal of adhering organic debris, which could harbor phosphatase-producing microbes, could easily damage ectomycorrhizal root tips. As excised root tips are deprived of a C flux from the tree following excision, measurements should be taken rapidly, within a few days after sampling and storage at 4 °C [67].

### 3.4. Assessing Differences between Guilds of Mycorrhizal Trees

Comparing spatially separated forests with either mycorrhizal guilds can be used to assess the effects of N deposition on P acquisition by EcM fungi, but differences in soil properties between these forest stands may override any mycorrhizal effect. Studies in areas with the same soil substrate and across a gradient of mycorrhizal symbiosis would therefore be preferable [37,53]. This approach requires a correct assessment of the relative abundance of both mycorrhizal tree species and mycorrhizal fungal groups. Unfortunately, studies by Fan et al. [54] and Hu et al. [57] are problematic in this respect. They assessed AM fungal abundance by a specific PLFA, which is also produced by bacteria [68], rather than by the specific NLFA that is only produced by AM fungi; and EcM fungal abundance by a PLFA, which is not specific for EcM fungi but is for all Ascomycota and Basidiomycota. For that reason, the conclusions of those papers should be treated cautiously.

### 4. Conclusions

Bogar et al. [16] speculated that under N deposition, the ability of EcM trees to negotiate trading relationships (C-for-N trade) with EcM fungi and hence increase selectivity may increase. The question is whether that ability would equally translate into an improved ability to invest in EcM fungi that allow a more favorable C-for-P trade. This review found very little evidence that under N enrichment, EcM fungi that have on average higher P efficiency and thereby prevent or delay N:P imbalances might be selected. However, methodological issues in some of these studies make it imperative to treat this conclusion cautiously. Considering the importance of avoiding N:P imbalances under N enrichment and the need to restore EcM forests that have suffered from long-term excess N loading [14], further research into this question is urgently required. Experiments in mixed AM and EcM forests along gradients of N and/or P availability provide ample opportunities to address this relevant question. The resistance of EcM forests towards invasion by AM plants seems limited, although the ability of EcM fungi to modify P speciation might retard competitive replacement, a topic that equally merits further study.

**Author Contributions:** T.W.K. and L.M.S. conceived the ideas, did the literature search, and wrote the manuscript. All authors have read and agreed to the published version of the manuscript.

**Funding:** This research received no external funding.

**Data Availability Statement:** No new data were created or analyzed in this study. Data sharing is not applicable to this article.

**Acknowledgments:** We thank three reviewers for constructive comments on an earlier version of the manuscript.

**Conflicts of Interest:** We declare no conflict of interest.

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
