# Peer review of "Do Ectomycorrhizal Trees Select Ectomycorrhizal Fungi That Enhance Phosphorus Uptake under Nitrogen Enrichment?"

_forests, doi:10.3390/f14030467_

Round 1

Reviewer 1 Report

General comments

The aim of manuscript, to contribute to the understanding if under high N deposition EcM trees can select EcM fungi that are both tolerant to high N availability and efficient in the acquisition of P from literature data, is very ambitious.  I am sure that authors did it the best they could. The only remaining question is whether the authors had at their disposal enough representative and comparable data, created by comparative methods. I will give two examples from the manuscript, from which, in my opinion, it follows that the authors had some problems with this.

On line 492-496 the authors themselves admit that “Unfortunately, studies by Fan et al. [54] and Hu et al. [57] are problematical in this respect. They assessed AM fungal abundance by a specific PLFA, which however is also produced by bacteria [67] rather than by the NLFA, which is only produced by AM fungi; and EcM fungal abundance by a PLFA that is not specific for EcM fungi but for all Ascomycota and Basidiomycota. For that reasons the conclusions of those papers should be treated cautiously.” There is a question, if such data, difficult to compare for AM and EcM fungal species, should not be included in the literature synthesis at all.

In the second case, the question arises whether the authors correctly evaluated in all cases the necessary conditions for selecting usable and comparable results for their review. Authors write on line 480-482: “As excised root tips are deprived of a carbon flux from the tree after excision, measurements should be taken rapidly, within a few days after sampling and storage at 4 °C.” However, according my experiences with bioassay studies with excised roots, it was necessary to analyse nutrient uptake up to 24 hours after sampling, to obtain usable results.

From the above follows my request to the authors of the manuscript to carry out one more critical revision of their text, in two directions: i) if necessary, to add more critical comments to the used literary data, as they did on the already mentioned lines 492-496 and ii) in the interpretation of the results of the analysis of literary data, please, assess the overall credibility of the individual conclusions reached, according to the quality of data found.

Formal comments

The entire manuscript is very carefully processed, only a few remaining typos need to be removed, eg. on line 80 or 495.

Author Response

Thank you for referring to the aim our ms as ‘very ambitious’. It is evident that the literature search indicated that, while the current knowledge would provide a negative answer to our research question, some data are problematical, either because of the applied methodology and / or because conclusions are not necessarily explained by the data but can have alternative explanations as well. We think that reviews have three different purposes: (i) indicate what is known with sufficient certainty; (ii) indicate cases where results have been overinterpreted or where there are methodological issues with published reports; and (iii) indicate the way forward. Our approach is in that sense similar to that by Karst et al. (Positive citation bias and overinterpreted results lead to misinformation on common mycorrhizal networks in forests. Nature ecology & Evolution 2023. https://doi.org/10.1038/s41559-023-01986-1]. Because we think it is important to also list studies where conclusions are problematical, we consider it is better to include them with a cautionary note about these methodological issues, rather than to omit them (and potentially invite criticism because we overlooked that paper). Any fair treatment of such publications implies that we one first mentions the reported results before commenting on these issues. We therefore added criticisms where we thought this is the case and we are not aware that we neglected to criticise other papers where criticism is necessary. We would be happy to learn about specific cases where we failed to apply our critical attitude; for the time being, however, we do not see any possibility for textual changes.

We agree that the time between excision of root tips and enzymatic measurements should be as short as possible. It technically possible, this should indeed be done within 24 hours. However, some studies that took place in remote areas (e.g., tropical rain forests) could not have been executed if this criterion was applied in a very strict sense. For that reason, we were a bit liberal (and tried to avoid being too harsh) for this methodology. We are seeing support for our somewhat liberal attitude by Pritsch et al. (Mycorrhiza 21: 589-600. 2011) who stated that “Based on the results of experiments 2–4, in which different ECM were stored in tap water, it appears that storing ectomycorrhizal tips in tap water ranging from overnight storage to 1 week at 4°C is possible to conveniently study enzyme activities.” It may well be that assessment of nutrient uptake demands shorter storage, but we focused on enzymatic assays in l. 409-412. We now added this reference to the manuscript.

With respect to the overall credibility of the results reported we had in our manuscript conclusions per section in boldface, which we thought was a fairly neutral way of summarising the data. If this boldface is lacking, we refrained from drawing strong conclusions.

Both typing errors have been corrected. Thank you.

Reviewer 2 Report

Authors tried to review the scientific literature available and have addressed the question,  Do ectomycorrhizal trees select ectomycorrhizal fungi that enhance phosphorus uptake under nitrogen enrichment? They have also highlighted the limitations and methodological issues associated with the study in question. It is a good piece of compiled information.

Minor errors:

Line 61: add space mycorrhizalguild

Line 598, 616, 619: remove highlight from doi

In references section dois are not uniformly formatted.

https:// is missing e.g., 525,527, 529, 537, 540, 553, 558 and many more while others do have https://. Please follow the journal style.

Author Response

Thanks, we corrected the typing errors; and now made a consistent lay out when referring to the doi

Reviewer 3 Report

Dear authors,

Could climate change, and global warming in particular, have an impact on the studied relationship between nitrogen and phosphorus in the soil? After periods of drought, which are becoming more frequent, there is a nitrogen deficiency and it is advisable to replenish it despite a constant supply of nitrogen, is it not? Such situations have been observed in oak stands during several drought years. Using reconnaissance types, ectomycorrhizae were classified according to ecologically relevant characteristics. The contact type was significantly correlated with C:N and Corg, while the short-distance type was correlated with Ca, phosphorus (P2O5) and pH. The mid-distance exploration type was significantly correlated with fine-grained soil particle size fractions: coarse silt (0.05-0.02 mm) and fine silt (0.02-0.002 mm) and clay (> 0.002 mm). The long distance type showed a similar pattern to the smooth medium distance type, but also correlated with nitrate (N) [doi:10.3390/f10010030 ]. Disturbance and local dieback of stands increase grass cover and thus nitrate content? Phosphorus uptake in turn depends on the availability of phosphorus to plants in the soil, which in turn depends on the acidity of the soil, does not it? In an excessively acidic environment, water-insoluble compounds such as iron phosphate form, do not they? On the other hand, microorganisms can help in the uptake of phosphorus compounds by forming soluble forms, but they limit the growth of mycorrhizae (doi:10.3390/f9100597). So do all these processes affect the proposed EcM model for improving phosphorus uptake under nitrogen enrichment? Moreover, mycorrhizae depend on the age and species of the trees, and species diversity is constantly changing, is not it? Some nurseries mycorrhage seedlings with EcM-type mycorrhizae, should they prefer the right ones... which ones? Does the disappearance of a tree species in a stand, e.g. the spruces currently being killed by the bark beetle Ips typographus, lead to the disappearance of the associated EcM species?

Author Response

Thank you for your questions. While they do not entail suggestions for modifications of the text per se, we have reflected on the responses and to the extent to which they should be added to the revised version.

We have indicated how climate change and nitrogen deposition may be connected (l. 395-410). Climate change (higher temperatures, at least under conditions where water does not become the main limitation factor) also affects decomposition and mineralisation (which equally affects N and P release), desorption processes (which is less sensitive to higher temperatures than mineralisation as it is mainly a physicochemical process, whereas mineralisation is mainly a biological process) and uptake of the released nutrients (which equally affects N and P). However, as our focus is on N deposition (and not on possible interactions between N deposition and climate change) we felt that expanding that paragraph would result in a less focused paper. Under conditions of drought, N fertilisation may NOT be a good idea – N uptake would be limited because of drought (as plants close their stomata to reduce water loss, they also reduce uptake); and when rainfall starts the available N might be higher than the uptake capacity, resulting in leaching (directly as nitrate; indirectly when ammonium excess is first oxidised to form nitrate) and simultaneous acidification. 

Thank you for the reference to the Bdzyk et al. paper (Forests 2019). Unfortunately, we did not see evidence that under higher N loads there were different associations with ectomycorrhizal fungi, as the paper contained data on total N (mainly organic N; the amounts of mineral N (ammonium and nitrate) are only a minor fraction of total N), which did not allow assessing the extent to which the three stands were differentially affected by N deposition. The (non-significant; to judge from reported standard errors) decrease in pH and increase in total C and N would point to reduced decomposition, which could have been caused by acidification rather than by N deposition. We therefore did not see a good place to refer to the paper in our manuscript.

Under more acidic conditions P availability indeed is lower. However, this is not due to precipitation as iron phosphates but a consequence of more positive charge on iron (hydr-)oxides that results in stronger sorption (and hence more complex desorption, see N.J. Barrow 2021. Comparing two theories about the nature of soil phosphate. Eur. J. Soil Sci. 72: 679-685. DOI: 10.1111/ejss.13027). Unfortunately, the importance of acidification-driven processes on P sorption and hence availability (which would have allowed our discussing the study) cannot be assessed by the method through which P was determined after citrate extraction as that desorbs part of the adsorbed P. This factor might also explain why P levels were not statistically significantly different in the three sites in Bdzyk et al. study.

There are many more processes that affect P uptake, but we have not found evidence that these processes are affected by N deposition. However, we do think that some of these would deserve further study.

Some other questions refer to oak decline as a general phenomenon, for which there is multiple causality (N deposition, acidification, drought, temperature, insect pests, novel pathogens). While it would be a major question whether under other processes of oak decline tree P nutrition is affected and whether tree have compensatory mechanisms through selective association with specific ectomycorrhizal fungi, we feel that, based on our current knowledge, such a section would be extremely speculative, which made us decide not to discuss the topic. But again, it is very worthy of investigation.